# Compass-model physics on the hyperhoneycomb lattice in the extreme spin-orbit regime

Ryutaro Okuma ®[1,2] ✉, Kylie MacFarquharson[1], Roger D. Johnson[3], David Voneshen ®[4,5], Pascal Manuel ®[4] & Radu Coldea ®[1]

The physics of spin-orbit entangled magnetic moments of $4d$ and $5d$ transition metal ions on a honeycomb lattice has been much explored in the search for unconventional magnetic orders or quantum spin liquids expected for compass spin models, where different bonds in the lattice favour different orientations for the magnetic moments. Realising such physics with rare-earth ions is a promising route to achieve exotic ground states in the extreme spin-orbit limit; however, this regime has remained experimentally largely unexplored due to major challenges in materials synthesis. Here we report the successful synthesis of powders and single crystals of $\beta$-$Na_2PrO_3$, with $4f^1$ $Pr^{4+}$ $j_{eff} = 1/2$ magnetic moments arranged on a hyperhoneycomb lattice with the same threefold coordination as the planar honeycomb. We find a strongly non-collinear magnetic order with highly dispersive gapped excitations that we argue arise from frustration between bond-dependent, anisotropic off-diagonal exchanges, a compass quantum spin model not explored experimentally so far. Our results show that rare-earth ions on threefold coordinated lattices offer a platform for the exploration of quantum compass spin models in the extreme spin-orbit regime, with qualitatively distinct physics from that of $4d$ and $5d$ Kitaev materials.

Materials with magnetic moments interacting via bond-dependent anisotropic interactions are attracting much attention as candidates to display novel cooperative behaviour. This is exemplified by the Kitaev model on the honeycomb lattice for spin-1/2 moments illustrated in Fig. 1a, where Ising exchanges $K S_i^{\gamma} S_j^{\gamma}$ with $\gamma = x, y, z$ couple mutually orthogonal spin components for the three bonds sharing a common site, with the resulting strong frustration stabilising an exactly solvable quantum spin liquid[1]. Candidates to realise such physics are heavy transition metal ions, such as $4d^5$ $Ru^{3+}$ or $5d^5$ $Ir^{4+}$ ions, located inside edge-sharing cubic octahedra, where the combination of spin-orbit coupling and crystal field stabilises magnetic moments with mixed spin-orbital character, that can interact via bond-dependent anisotropic exchanges of predominant Kitaev character[2]. Experimental

studies of candidate materials[3] have revealed novel phenomena such as spin-momentum locking in $Na_2IrO_3$[4], unconventional continuum of excitations[5] and thermal transport[6,7] in $\alpha$-$RuCl_3$, and counter-rotating incommensurate orders in $\alpha$-, $\beta$- and $\gamma$-$Li_2IrO_3$[8–10].

An equally important yet distinct bond-dependent anisotropic interaction is the off-diagonal symmetric exchange $\Gamma$[11], which couples spin components normal to the Kitaev axes, i.e., $\Gamma(S_i^x S_j^y + S_i^y S_j^x)$ for a $z$-bond. Such terms also generate frustration as each spin is conflicted into pointing along incompatible directions favoured by the three bonds sharing that site, see Fig. 1b, with the set of directions changing upon reversing the sign of $\Gamma$ (see Fig. 1c), with both sets of directions different from those favoured by a Kitaev term as illustrated in Fig. 1a. The $\Gamma$ model on the honeycomb lattice has a macroscopically

[1]Clarendon Laboratory, University of Oxford Physics Department, Oxford OX1 3PU, UK. [2]Institute for Solid State Physics, University of Tokyo, Kashiwa Chiba 277-8581, Japan. [3]Department of Physics and Astronomy, University College London, London WC1E 6BT, UK. [4]ISIS Facility, Rutherford Appleton Laboratory, Chilton Didcot OX11 0QX, UK. [5]Department of Physics, Royal Holloway University of London, Egham TW20 0EX, UK. ✉e-mail: ryutaro.okuma@physics.ox.ac.uk

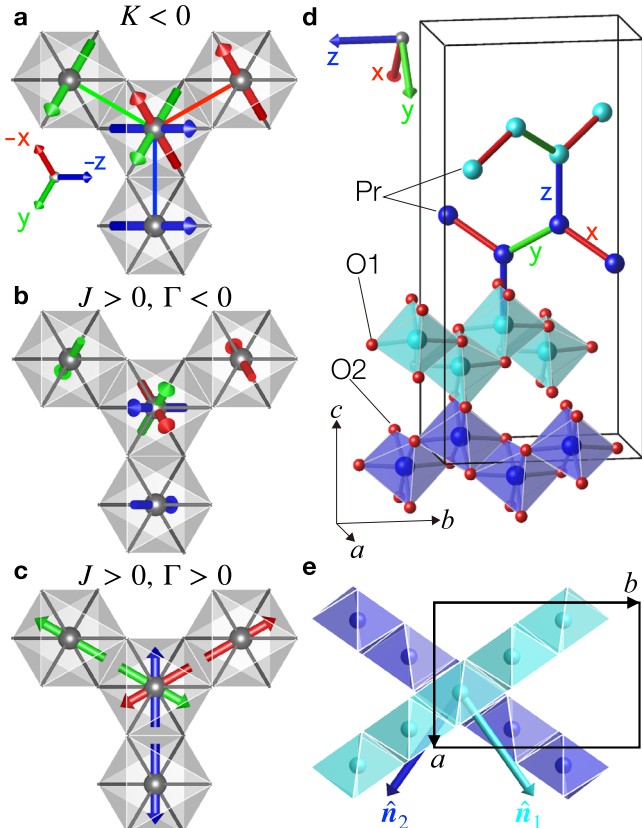

**Fig. 1 | Frustration from different bond-dependent anisotropic exchanges on the hyperhoneycomb lattice. a** Local threefold coordination of $Pr^{4+}$ ions (grey balls) inside edge-sharing cubic $O_6$ octahedra (grey shading). Spin pairs (arrows) for each bond are colour-coded red/green/blue according to the label xyz of the axis normal to the $Pr-O_2-Pr$ superexchange plane of that bond. In the Kitaev model with $K < 0$, each bond prefers the spins at the two ends to point along the direction normal to the superexchange plane, so the central spin is frustrated among the three orthogonal xyz directions. **b, c** The Γ interaction also creates frustration, but among other spin directions. **b** $J/\Gamma$ model for Heisenberg exchange $J > 0$ and $\Gamma < 0$. Each bond prefers spins at the two ends antiparallel, in the superexchange plane and orthogonal to the bond direction. The central spin is frustrated among three non-coplanar directions at 60° relative to each other. **c** Same as (**b**), but for $\Gamma > 0$, where each bond prefers spins at the two ends to be antiparallel and along the bond direction. The central spin is frustrated among three coplanar directions at 60° relative to each other. **d** Crystal structure of $\beta$-$Na_2PrO_3$ (orthorhombic *Fddd* space group, see Supplementary Note 2) showing the threefold coordinated hyperhoneycomb lattice of $Pr^{4+}$ ions (dark/light blue balls) located inside edge-sharing $O_6$ octahedra forming zigzag chains that alternate in direction between the two basal plane diagonals $\mathbf{a} \pm \mathbf{b}$ (Na ions not shown for brevity). Bonds are colour-coded and labelled as in (**a**) for an ideal structure with cubic octahedra. **e** Light/dark octahedral zigzag chains in (**d**) projected onto the basal *ab* plane. Coloured arrows labelled $\hat{\mathbf{n}}_{1,2}$ show normals to the distinct planes defined by the light/dark zigzag chains. This non-coplanarity enhances the frustration effects of Γ interactions.

degenerate manifold of classical ground states[12] and the quantum model is not exactly solvable. Symmetry-protected topological phases have been predicted in a honeycomb ladder with Γ and Heisenberg exchange[13].

Here we report experimental studies that find evidence for substantial Γ interactions in $\beta$-$Na_2PrO_3$[14] where $Pr^{4+}$ $4f^1$ ions form a three-dimensional hyperhoneycomb lattice with the same local threefold coordination as the planar honeycomb. $Pr^{4+}$ ions have long been theoretically predicted to realise quantum compass spin models with potentially different Hamiltonians compared to the heavy transition metal ions $Ru^{3+}$ and $Ir^{4+}$ due to the stronger spin-orbit coupling and the

different characteristics of the orbitals involved in superexchange[15,16]. However, up to now, no physical properties have been reported for $\beta$-$Na_2PrO_3$ because the synthesis is hampered by severe air-sensitivity and the presence of more stable polymorphs[17,18].

The three-dimensional hyperhoneycomb lattice is an ideal playground to explore frustration effects from Γ interactions. It has an orthorhombic unit cell with zigzag chains running alternately along the $\mathbf{a} \pm \mathbf{b}$ basal plane directions with vertical bonds connecting the two types of chains (see Fig. 1d) such that each site has three nearest neighbours. This magnetic lattice has been realised experimentally so far only in $\beta$-$Li_2IrO_3$[19] and $\beta$-$ZnIrO_3$[20], in the latter case with chemical disorder on the Zn site. In the absence of a Γ term, there is no critical difference between the planar honeycomb and the hyperhoneycomb: the Kitaev model on both lattices has exactly solvable quantum spin liquid ground states[21], and four types of similar collinear magnetic structures appear at the same value of $K/J$ at the mean-field level[11,22,23], where $J$ is the Heisenberg exchange. The introduction of Γ for the hyperhoneycomb lattice renders most of the phases non-collinear and even non-coplanar[23], whereas non-collinear orders are realised in the planar honeycomb only when $K$ and Γ are dominant[11]. The key difference is attributed to the fact that in the hyperhoneycomb structure, the zigzag chains are contained within distinct planes, as illustrated in Fig. 1e. This intrinsic non-coplanarity enhances the frustration effects from Γ interactions as one cannot define a global, common plane for all the spins, unlike the case of the two-dimensional planar honeycomb where all bonds are coplanar. Furthermore, in the coplanar case, a trigonal compression along the direction (Z) normal to the honeycomb layer can lead to bond-independent XXZ-type interactions, as in the case of the Co-based honeycomb $BaCo_2(AsO_4)_2$[24]. An Ising-like XXZ model has also been proposed to describe the layered honeycomb $\alpha$-$Na_2PrO_3$[18]. In contrast, such a model is not applicable to the hyperhoneycomb lattice due to its twisted 3D connectivity, with any anisotropy originating instead from bond-dependent anisotropic exchanges.

## Findings of this study

We have succeeded in the selective synthesis of phase-pure powders and sizeable single crystals of $\beta$-$Na_2PrO_3$, which realises a hyperhoneycomb lattice with $j_{eff} = 1/2$ $Pr^{4+}$ magnetic moments. A critical insight that enabled the synthesis was understanding the role played by the melting species $Na_2O_2$ and NaOH in the chemical stability of the various polymorphs of $Na_2PrO_3$, see Fig. 2. By combining neutron diffraction and inelastic scattering with magnetic symmetry analysis and spin-wave calculations, we obtain a full solution of a highly non-collinear magnetic structure with gapped and strongly dispersive spin-wave excitations. We provide evidence that this physics is governed by frustrated bond-dependent anisotropic interactions but of a different character from the much-explored Kitaev exchange.

## Results

### Magnetic susceptibility

We first characterise the magnetic behaviour using powder magnetic susceptibility measurements plotted in Fig. 3a. The data can be fitted in the region of 20 to 300 K by a Curie-Weiss law $\chi(T) = \chi_0 + \mu_0\mu_{eff}^2/3k_B(T - T_{CW})$ with $\chi_0 = 6.12(9) \times 10^{-4} cm^3 mol^{-1}$, $T_{CW} = -15(1)$ K indicating overall antiferromagnetic interactions, and $\mu_{eff} = 0.81(1)\mu_B/Pr$, which implies a g-factor $g = 0.94(1)$, smaller than the 10/7 value expected in the limit of very weak cubic crystal field[25]. Such moment reduction was also observed for $\alpha$-$Na_2PrO_3$[18] and attributed to mixing with higher crystal-field levels. Figure 3a (inset) shows a clear anomaly at $T_N = 5.2$ K, attributed to the onset of magnetic order. This is more clearly seen in susceptibility measurements on single crystals in Fig. 3b, where a sudden decrease in the susceptibility along the orthorhombic *a*-axis is observed below the same temperature as in the powder data, as expected for the onset of a magnetic structure with dominant antiferromagnetic *a*-axis components. Heat capacity (Fig. 3b

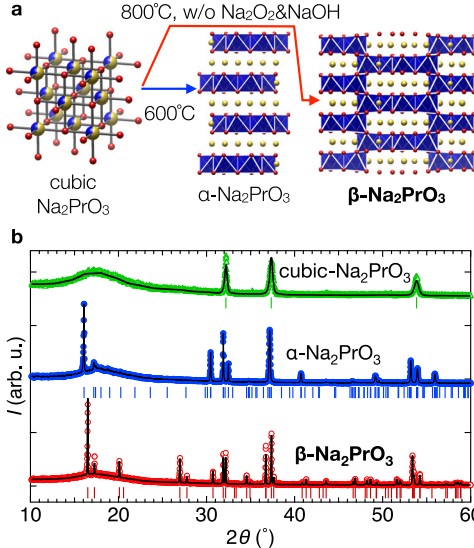

**Fig. 2 | Synthesis of hyperhoneycomb $\beta$-Na$_2$PrO$_3$. a** Phase pure cubic-Na$_2$PrO$_3$ synthesised at 400 °C is further annealed at 800 °C to obtain phase pure $\beta$-Na$_2$PrO$_3$. Annealing at a lower temperature of 600 °C stabilises $\alpha$-Na$_2$PrO$_3$. Red, blue, and yellow spheres indicate oxygen, praseodymium, and sodium atoms, respectively. **b** XRD powder diffraction patterns (monochromatic Cu-K$_{\alpha 1}$) of the three polymorphs of Na$_2$PrO$_3$, offset vertically for clarity. Red, blue and green symbols show the diffraction pattern of $\beta$-, $\alpha$-, and cubic-Na$_2$PrO$_3$, respectively. Black lines and bars under each data set indicate Rietveld refinement fits and positions of Bragg peaks for each phase, $\alpha$- from ref. [40], $\beta$- from Supplementary Table I, and cubic-phase from ref. [41].

lower trace) further corroborates this scenario by observing a very sharp peak at the same temperature as the kink in susceptibility.

## Magnetic propagation vector

To determine the magnetic structure, we use neutron powder diffraction, which revealed new diffraction peaks as well as an intensity increase at some structural peak positions upon cooling below $T_N$. The temperature dependence of the diffraction intensity is illustrated in Fig. 3c, which shows the intensity at a position where the structural Bragg diffraction is almost absent (as explained in Supplementary Note 3). At base temperature, a clear peak is observed, which decreases monotonically upon heating and can no longer be detected at $T_N$. The magnetic peaks are most clearly revealed in the difference pattern between base temperature (1.4 K) and paramagnetic (10 K), shown in panels d–f, where green bars under the pattern indicate the nominal peak positions for the F-centred lattice. Fig. 3d reveals a structurally forbidden peak at (002) around a $d$-spacing of 10 Å, which breaks the selection rule for (00$l$) structural reflections with $l = 4p$ ($p$ integer) characteristic of the structural $Fddd$ space group (due to the $d$ diamond glides normal to the $a$ and $b$ axes). Over 10 magnetic Bragg peaks could be detected in total, and all could be indexed by all-odd or all-even Miller indices, indicating that the magnetic and structural unit cells are the same, i.e., the magnetic propagation vector is **q** = **0**.

## Magnetic basis vectors

Symmetry analysis showed that any given magnetic structure can be decomposed into a linear combination of 12 modes: $F_i$, $A_i$, $C_i$, and $G_i$, where $i$ denotes the polarisation of the mode (along $xyz$ axes, defined to be along the orthorhombic $abc$ axes) and $F$, $A$, $C$ and $G$ denote basis

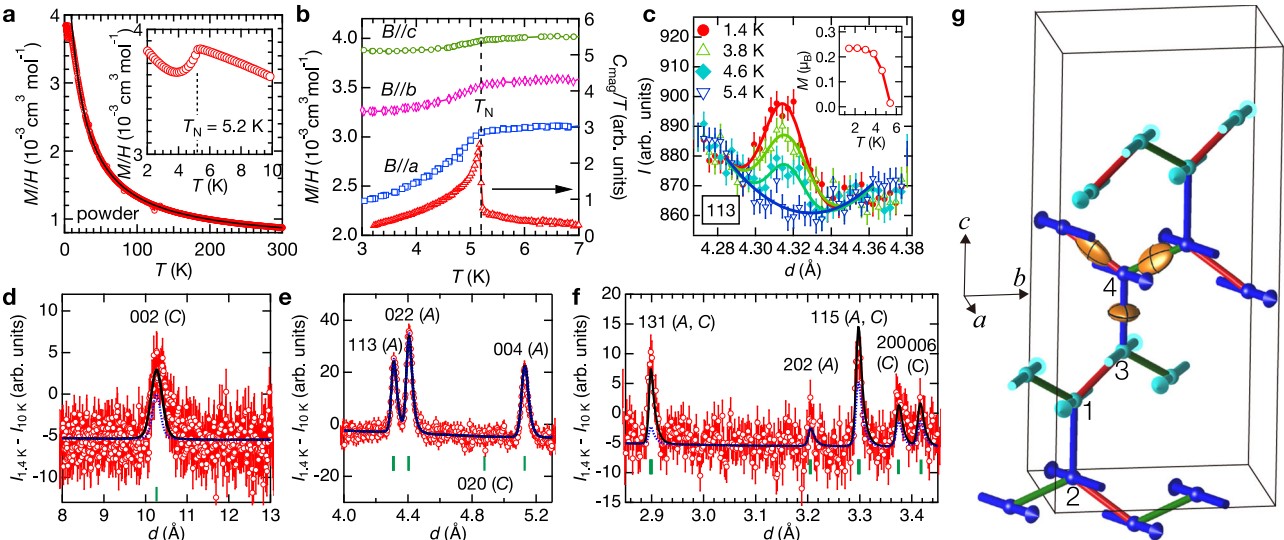

**Fig. 3 | Non-collinear magnetic order in $\beta$-Na$_2$PrO$_3$. a** Temperature dependence of the magnetic susceptibility (red circles) under field-cooling (FC) of 1 T. No difference is observed between FC and zero-FC. The black line is a fit to a Curie-Weiss form in the region 20 to 300 K. (Inset) Susceptibility anomaly near the onset of magnetic ordering. **b** Temperature dependence of single-crystal susceptibility and ac heat capacity of a multi-crystal sample. Blue squares, pink diamonds, green circles and red triangles indicate the susceptibility in field applied along the $a$, $b$ and $c$ axes and zero-field heat capacity (right-hand axis), respectively. The heat capacity points are raw data with an estimate of the addenda contribution subtracted. **c** Temperature dependence of the raw neutron diffraction intensities near the magnetic Bragg peak position (113) with an almost absent nuclear contribution, solid lines are fits to Gaussian peaks with a second order polynomial as a background determined from a fit to the data at 5.2 K. Inset shows the temperature dependence of the fitted magnitude of the ordered magnetic moment (solid line is guide to the eye). **d** Magnetic Bragg peak at the nuclear forbidden position (002) observed in the bank that covers the largest $d$-spacing. Data points are the intensity

difference between base temperature (1.4 K) and paramagnetic (10 K), and the green vertical bar under the pattern shows the nominal magnetic Bragg peak position. Solid black/dotted blue lines are fits to the ($A_x$, $\mp C_y$) models described in the text. Symbols and lines have the same meaning in panels (**e**) and (**f**), which show magnetic reflections with lower $d$-spacing. Error bars on data points in panels (**c**–**f**) represent one standard deviation. **g** Schematic diagram of the strongly non-collinear magnetic structure ($A_x$, $-C_y$) refined from neutron powder diffraction data (for projections onto different crystallographic planes, see Supplementary Fig. 6). Labels 1–4 indicate sites equivalent to those in the primitive cell (listed in Supplementary Table VI) up to $F$-centring translations. The anisotropy of the bond-dependent interactions is illustrated by the yellow ellipsoids with principal planes indicated by black contours. For the zigzag bonds the ellipsoids are elongated along the bonds, on the vertical bonds they are elongated transverse to the bonds, along the $a$-axis with the largest antiferromagnetic ordered components. For better visualisation of the ellipsoid's shape, we used $\Gamma/J$ and $\Gamma'/J$ twice as large as the estimated values from fitting the magnetic excitation spectrum.

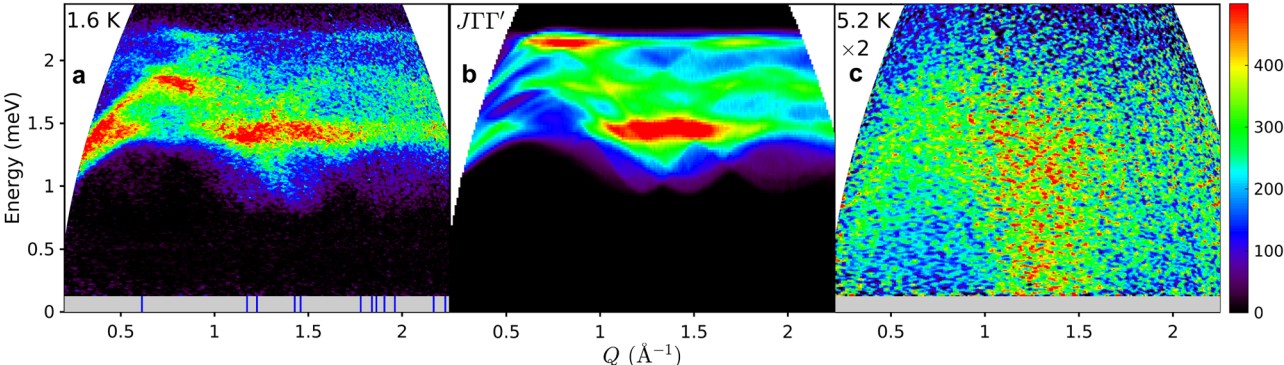

**Fig. 4 | Powder inelastic neutron scattering spectrum. a** Base temperature data (1.6 K, $E_i$ = 3.7 meV) observing dispersive magnetic excitations above a spin gap. Thin vertical blue bars at the bottom of the panel indicate wavevectors of magnetic Bragg peaks of the determined magnetic structure. Grey shading shows the inaccessible region close to the elastic line dominated by incoherent elastic scattering. **b** Spherically averaged spin-wave spectrum of the minimal $J\Gamma\Gamma'$ model described in the text, convolved with the estimated experimental energy resolution. **c** Same as (**a**), but at 5.2 K and with intensities scaled by ×2. The enhanced inelastic signal in a broad momentum range near 1.3 Å⁻¹ extending down to the lowest energies is attributed to precursor dynamical correlations from which the magnetic order develops upon cooling. The colour bar indicates scattering intensity in arbitrary units on a linear scale.

vectors, which encapsulate symmetry-imposed relations between the moment orientations (parallel or antiparallel) at the four Pr sites in the primitive cell as listed in Supplementary Table VI. *F* means ferromagnetic, *A* nearest-neighbour antiferromagnetic, *C* has parallel (antiparallel) spins on the vertical (zigzag) bonds, and *G* vice versa. Each basis vector satisfies distinct selection rules (summarised in Supplementary Table VII) with the consequence that simply the presence of certain magnetic Bragg peaks uniquely identifies the presence of certain basis vectors. In addition, the fact that magnetic neutron scattering is only sensitive to the magnetic moment components perpendicular to the wavevector transfer provides further information to identify the polarisation of the basis vectors. Finite intensity for the pure *C*-peaks (006) and (200) in Fig. 3f and absence of a peak at the pure *C* position (020) in Fig. 3e means that a basis vector *C* must be present and it must be polarised along the *y*-axis. Finite intensity at the mixed *AF* position (004) in Fig. 3e identifies the second basis vector as $A_x$, since $A_y$ would lead to the unphysical situation of unequal moment magnitudes on the four sites in the primitive cell, and any *F* (ferromagnetic) component can be ruled out by the absence of a ferromagnetic anomaly at $T_N$ in the magnetisation data in Fig. 3a (inset). In the following, we show that the basis vectors $A_x$ and $C_y$ found by inspection of the neutron powder diffraction pattern can describe it fully quantitatively, resulting in a complete solution of the magnetic structure.

### Full magnetic structure

A magnetic structure with only $A_x$ and $C_y$ basis vectors corresponds to a single irreducible representation (irrep), $m\Gamma_4^-$, for symmetry-allowed **q** = **0** magnetic structures (tabulated in Supplementary Table VIII), consistent with a continuous transition at $T_N$, as observed in the heat capacity data in Fig. 3b. The magnitudes $M_x$ and $M_y$ of the two basis vectors can be separately determined from the intensity of magnetic Bragg peaks where only one of them contributes, such as (004) for $A_x$ and (002) for $C_y$. The relative phase between the two basis vectors can be determined from the intensity of magnetic Bragg peaks where both basis vectors contribute, as the total intensity is the sum of the intensities due to the two separate basis vectors, plus an additional cross-term that is sensitive to the relative phase (for details see Supplementary Note 2). The magnetic diffraction pattern contains many magnetic Bragg peaks that can be used for this purpose, all peaks with *hkl* all-odd have contributions from both *A* and *C* basis vectors. Symmetry analysis predicts that the two basis vectors can be in-phase ($A_x$, $C_y$), or in antiphase ($A_x$, $-C_y$), and these two scenarios can be differentiated by neutron diffraction, as we will show later. The in-phase/

antiphase structures have ordered spins oriented predominantly perpendicular (parallel) to the direction of the zigzag chains, with the exact orientation being a degree of freedom related to the relative magnitudes of $M_x$ and $M_y$.

We have tested both scenarios, freely refining $M_x$ and $M_y$ in the fits. The best fit to an antiphase ($A_x$, $-C_y$) structure gives a very good account of the observed pattern, all peaks in Fig. 3d–f panels are quantitatively accounted for (black solid line, for more details of the fits see Supplementary Note 2). In contrast, the alternative fit (dashed blue) to an in-phase ($A_x$, $C_y$) structure cannot fully account for the intensity at well-measured peaks such as (002) in Fig. 3d and (131), (115) in panel f. In the best fit structure, the total ordered moment at each site is $M = \sqrt{M_x^2 + M_y^2} = 0.222(4)\mu_B$ with the relative ratio $M_y/M_x = 0.55(2)$. The resulting magnetic structure has spins confined to the *ab* plane, close (at an angle of 26(1)°) to the direction of the zigzag chains that they belong to. The resulting global magnetic structure is strongly non-collinear, as moments belonging to zigzag chains connected by a vertical bond make a relatively large angle $180° - 2\arctan(M_y/M_x) = 123(2)°$, see Fig. 3g. The refined magnetic structure naturally explains the dominant features in the temperature-dependent susceptibility data in Fig. 3b, which observed a prominent suppression below $T_N$ of the susceptibility along the *a*-axis, the direction with the dominant antiferromagnetic components, contrasting with an almost constant susceptibility below $T_N$ for field along the *c*-axis, normal to the plane of the ordered spins.

### Inelastic neutron scattering

To gain insight into the interactions that could stabilise the determined non-collinear structure, we performed powder inelastic neutron scattering measurements. The observed magnetic excitation spectrum deep in the ordered state is shown in Fig. 4a as a function of energy and wavevector transfer *Q*. The spectrum has a clear gap of around 0.75 meV and extends up to 2.3 meV, being highly structured with a combination of highly dispersive features and prominent near-flat regions. As expected, the energy scale of the spectrum is comparable with the Curie-Weiss temperature of $-k_B T_{CW} \simeq 1.3$ meV. The lower boundary of the spectrum is highly dispersive with clear gapped minima seen for *Q* near 1.25 and 1.45 Å⁻¹, which are close to wavevector positions where several magnetic Bragg peaks occur in the diffraction pattern, indicated by thin vertical blue bars at the bottom of the panel. The magnetic character of the observed spectrum is further confirmed by measurements at $T_N$ shown in Fig. 4c, the gaps fill in and the dispersive and strong intensity features observed at lower temperature wash out.

## Spin gap

The substantial spin gap indicates a considerable energy cost in moving the spins away from their local orientations in the magnetic ground state. The presence of such energetically strongly preferred directions cannot be due to single-ion anisotropy effects. For a Kramers ground state doublet as expected for $4f^1$ $Pr^{4+}$ ions in octahedral crystal field[16], there cannot be any local, single-ion anisotropy terms, as all even powers of the components of the effective angular momentum of 1/2 describing the ground state doublet are constants[26]. The substantial gap observed therefore must be due to anisotropic exchange interactions, as we discuss below.

## Spin Hamiltonian

The non-collinear magnetic order observed could, in principle, be stabilised by the competition between just two exchange terms, a nearest-neighbour antiferromagnetic Heisenberg exchange $J$ and a symmetry-allowed Dzyaloshinskii-Moriya interaction $D$ on the vertical bonds. However, to quantitatively explain the large angle between moments in the $ab$ plane would require an unphysically large $D/J$ ratio and also such a model would have a gapless spectrum, in contrast with the substantial gap seen experimentally (for more details, see Supplementary Note 5). We therefore consider the origin of the observed magnetic structure and spin-wave excitations within a $JK\Gamma$ model, expected microscopically via a strong-coupling expansion of a multiband tight-binding model with on-site interaction and large atomic spin-orbit coupling, in this framework $K$ and $\Gamma$ appear in the presence of a Hund's coupling[16,23]. Such a model with $J > 0$, $K = 0$ and $\Gamma < 0$ can qualitatively reproduce several of the key features observed experimentally. First, it gives a non-collinear magnetic ground state with moments confined in the $ab$ plane, with the same basis vectors $A_x$ and $C_y$ found experimentally, and with $A_x$ dominant. However, the structure is predicted to be the in-phase combination $(A_x, C_y)$ (labelled $AF_a$ in ref. 23), not the antiphase $(A_x, -C_y)$ found experimentally. Second, it naturally gives a spin gap, scaling as $\sqrt{-\Gamma J}$ at leading order in $\Gamma/J$.

The antiphase basis vector combination $(A_x, -C_y)$ with $A_x$ dominant is not contained in the phase diagram of the above $JK\Gamma$ model[23], however it can be stabilised by relaxing the assumption that all bonds are symmetry-related. Indeed, in the $Fddd$ crystal structure, the zigzag bonds are not symmetry-equivalent to the vertical bonds. By calling the off-diagonal exchange on the zigzag bonds $\Gamma'$ and making $\Gamma' > 0$ on the x-bond of the chains running along $\mathbf{a} - \mathbf{b}$ (such as 1–3 in Fig. 3g) stabilises the basis vector combination $(A_x, -C_y)$ as found experimentally. This suggests that the symmetry-inequivalence of the zigzag and vertical bonds is the most likely reason for the observed antiphase basis vector combination in the ground state.

To illustrate the level of agreement that can be obtained by such a minimal model, we assume for simplicity equal magnitude and opposite sign off-diagonal exchanges on the zigzag and vertical bonds, i.e., $\Gamma' = -\Gamma$ and perform a fit of the observed powder-averaged spectrum freely varying $J$ and $\Gamma$, this gives $J = 1.22(4)$ meV and $\Gamma = -0.27(5)$ meV. This minimal model reproduces reasonably quantitatively the dominant highly dispersive features and regions of strong intensity in the powder-averaged spectrum, compare Fig. 4a with b (for further details of the mean-field analysis and spin-wave calculations see Supplementary Notes 4–6). We take this agreement as indication that the bond-dependent $\Gamma$ and $\Gamma'$ interactions are the most relevant subleading exchanges after the isotropic Heisenberg term $J$. In the above minimal model, the mean-field calculated $M_y/M_x = 0.14$ (assuming an isotropic $g$-tensor in the $ab$ plane) is smaller than the ratio 0.55 deduced experimentally, also there are differences of detail between the observed and calculated powder spectrum in Fig. 4a and b, we attribute those differences to extensions of the Hamiltonian beyond the minimal model considered here, which however we do not expect would change the physics qualitatively, but could improve the level of quantitative agreement with the experiment.

## Ground state selection

In the minimal model considered above, the ground state selection occurs as follows. An antiferromagnetic Heisenberg exchange $J > 0$ on all bonds selects collinear Néel order, adding off-diagonal exchange $\Gamma < 0$ on the vertical bonds breaks the spherical rotational symmetry, opens a gap in the spectrum and selects the $a$-axis for the moments' direction in the ground state (basis vector $A_x$). Adding now off-diagonal exchange $\Gamma' > 0$ on the zigzag bonds rotates the moments at the two ends of a vertical bond in opposite senses, but keeps all spins in the same zigzag chain collinear and antiferromagnetically ordered by mixing in a $-C_y$ basis vector in the ground state; this is such as to bring the spins towards locally favoured easy directions in each chain, with those directions different between chains connected by a vertical bond and related by a $2_x$ rotation. Therefore, the non-collinearity of the magnetic order can be understood as a direct consequence of the frustration between the off-diagonal exchanges on the different bonds.

# Discussion

The magnetic behaviour of $\beta$-Na$_2$PrO$_3$ is quite different from that of isostructural $\beta$-Li$_2$IrO$_3$[19], the only other known material with a magnetic hyperhoneycomb lattice with no structural disorder. The latter material has a non-coplanar, incommensurate magnetic structure with counter-rotating moments[9], in contrast $\beta$-Na$_2$PrO$_3$ has a non-collinear, commensurate magnetic structure. The underlying spin Hamiltonians are qualitatively different, in $\beta$-Li$_2$IrO$_3$ a dominant $K$ has been proposed[27], whereas in $\beta$-Na$_2$PrO$_3$ we find a $J\Gamma\Gamma'$ model, we attribute this difference to the distinct orbitals and superexchange mechanisms involved in the two cases. The presence of a clear spin gap in $\beta$-Na$_2$PrO$_3$ of magnitude comparable to the Zeeman energy of accessible applied magnetic fields opens up the prospect of observing experimentally novel field-induced magnetic phases and unconventional spin dynamics, such as topological nodal lines and Weyl magnons, protected by magnetic glide symmetries and arising from magnon pairing effects, theoretically predicted[28] for generic bond-dependent anisotropic Hamiltonians on the hyperhoneycomb lattice, but not yet observed experimentally.

By incorporating rare-earth $4f$ ions in threefold coordinated lattices, with orbitals of different character mediating the superexchange interactions in the extreme spin-orbit regime, we have been able to access a different materials platform for realising quantum compass spin models with largely distinct physics from the much explored $4d$ and $5d$ Kitaev materials. We have facilitated the first steps by resolving materials synthesis challenges and observed new physics driven by frustrated off-diagonal exchanges.

Beyond $\beta$-,$\gamma$-Li$_2$IrO$_3$ and $\beta$-Na$_2$PrO$_3$, realisation of three-dimensional threefold coordinated lattices in more systems will be an important task to reveal the rich physics of the quantum compass models. Notable progress in this direction is the recently demonstrated control of cation ordering in the rock salt structure, used to obtain a three-dimensional network of corner- and edge-sharing octahedra in Li$_3$Co$_2$SbO$_6$[29,30]. The use of high pressure could also potentially transform layered honeycomb materials into 3D hyperhoneycomb lattices, as illustrated by high-pressure studies on IrI$_3$[31]. Very recently, an organic molecule was employed to realise another threefold coordinated lattice of much theoretical interest, the hyperoctagon lattice, in a Co-based metal organic framework[32]. We anticipate that these and other novel synthetic procedures will expand the materials platform of three-dimensional threefold coordinated lattices and allow a wider experimental exploration of the rich range of cooperative magnetic behaviours expected for such geometries in the

presence of strong spin-orbit coupling, as we have revealed in hyper-honeycomb $\beta$-$Na_2PrO_3$.

## Methods

### Synthesis
Full details of the synthesis using a solid-state reaction protocol under inert atmosphere are provided in Supplementary Note 1.

### Crystal structure determination
The $\beta$-phase crystal structure was determined via single-crystal x-ray diffraction using a Mo source SuperNova diffractometer on a crystal with dimensions $170 \times 95 \times 31\,\mu m^3$, covered with vacuum grease to protect it from air. No evidence for sample degradation was observed within the duration of the x-ray measurements (less than a couple of hours).

### Magnetic characterisation
Magnetisation measurements were performed using a Quantum Design MPMS3 system in fields up to 7 T and temperatures down to 2 K, first on powders of typical mass 22.7 mg and subsequently on co-aligned single crystals with a total combined mass of order 0.2 mg. The single crystals were initially handled in vacuum pump oil, which was removed by washing with toluene. For the magnetisation measurements in field along specific crystallographic directions ($a$, $b$, $c$) the crystals were aligned and fixed onto a flat plate (single crystal of NaCl with typical dimensions $1.4 \times 1.3 \times 0.7\,mm^3$). Melted paraffin wax was used to coat the crystals and fix them onto the flat plate and an aluminium foil was attached below the flat plate to almost cancel the diamagnetic contribution of paraffin wax. After the measurements along all the axes, the crystals were removed by melting the wax and the background susceptibility signal, comprising of diamagnetism from wax and NaCl, and paramagnetism of the aluminium foil, was measured and subtracted off to obtain the intrinsic $Na_2PrO_3$ susceptibility. Both the powder and single-crystal susceptibility measurements showed evidence for a small ferromagnetic impurity, identified as $PrO_{2-x}$ due to decomposition. The contribution of this impurity was subtracted by comparing the magnetisation data to single-crystal torque data (to be described in detail elsewhere).

### Heat capacity
Zero-field heat capacity was performed on a collection of single crystals of combined mass 0.12(2) mg using a custom ac heat capacity setup operating with a Quantum Design PPMS system.

### Neutron diffraction
Neutron powder diffraction measurements were performed using the time-of-flight diffractometer WISH at the ISIS Facility in the UK. The sample was a 15 g powder of $\beta$-$Na_2PrO_3$ loaded in a thin-walled aluminium can. The powder contained a small amount of NaOH impurity, which resulted in an increased background signal. Diffraction patterns were collected at base temperature (1.4 K) and paramagnetic (10 K) for about 8 h each at an average proton current of 30 µA, with additional data collected at a selection of intermediate temperatures to obtain the order parameter. The raw time-of-flight neutron data were normalised and converted to $d$-spacing using the MANTID[33] package. Rietveld refinements of crystal and magnetic structure models were performed using FullProf[34], simultaneously against data measured in detector banks 1 to 10 (grouped in pairs). A small absorption correction was included in the refinements to account for moderate neutron absorption by Na. The result of the structural refinement is presented in Supplementary Fig. 4.

### Neutron spectroscopy
Inelastic neutron scattering measurements were performed using the direct-geometry time-of-flight neutron spectrometer LET, also at ISIS.

The sample was the same as for the powder neutron diffraction measurements described above. Most data was collected with LET operated in repetition rate multiplication mode to measure the inelastic scattering of neutrons with incident energies of $E_i$ = 3.7, 7.5 and 22.5 meV. The raw time-of-flight neutron data were corrected for detector efficiency and converted to intensity as a function of momentum transfer and energy $S(Q, \omega)$ using the MANTID[33] package. The data in Fig. 4a was counted for a total of 9 h at an average proton current of 40 µA. At higher energy transfers, the INS data showed visible non-dispersive inelastic peaks near 3 and 7.3 meV, attributed to well-known[35] transitions between crystal-field levels of $Pr^{3+}$ ions in $Pr_6O_{11}$, present in the powder sample as a small impurity phase. This high-energy data was excluded from the analysis, at base temperature this signal is well isolated from the lower energy and strongly dispersive signal in Fig. 4a attributed to cooperative magnetic excitations in the primary $\beta$-$Na_2PrO_3$ phase. Calculations of the spin-wave spectrum for model Hamiltonians were performed in the primitive cell with four magnetic sublattices using SpinW[36], for more details see Supplementary Notes 3–5. The spherically averaged spin-wave spectrum was then compared to the experimentally measured $S(Q, \omega)$ to obtain the best fit Hamiltonian parameters.

## Data availability
The experimental data supporting this research is openly available from ref. 37.

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

## Acknowledgements

We acknowledge support from the European Research Council under the European Union's Horizon 2020 research and innovation programme Grant Agreement Number 788814 (EQFT) (R.O., K.M., R.C.) and the Engineering and Physical Sciences Research Council (EPSRC) under grant No. EP/M020517/1 (R.C.). R.O. acknowledges support from JSPS KAKENHI (Grant No. 23K19027JST) and JST ASPIRE (Grant No. JPMJAP2314). R.C. acknowledges support from the National Science Foundation under Grants No. NSF PHY-1748958 and PHY-2309135, and hospitality from the Kavli Institute for Theoretical Physics (KITP) where part of this work was completed. The neutron scattering measurements at the ISIS Facility were supported by beamtime allocations from the Science and Technology Facilities Council[38,39]. We thank Andrew Boothroyd for sharing his determination of the spherical magnetic form factor of $Pr^{4+}$ and for pointing out ref. 35 with crystal-field transitions in $Pr_6O_{11}$. For the purpose of Open Access, the authors have applied a CC BY public copyright licence to any Author Accepted Manuscript version arising from this submission.

## Author contributions

R.C., R.D.J. and R.O. conceived research; R.O. developed the synthesis protocol for powders and single crystals and performed structural and magnetic characterisation; K.M. performed single-crystal heat capacity and torque experiments. R.O., P.M., R.D.J. and R.C. performed neutron powder diffraction measurements, and R.O. analysed this data to solve the magnetic structure. R.O., D.V. and R.C. performed inelastic neutron scattering measurements; R.C. and R.O. analysed this data and performed theoretical calculations. R.O., R.C. and R.D.J. wrote the paper and the supplementary information with input from all co-authors. R.C. supervised all aspects of the project.

## Competing interests

The authors declare no competing interests.
