## [Peer Review file · Nature Communications]

Compass-model physics on the hyperhoneycomb lattice in the extreme spin-orbit regime

Corresponding Author: Dr Ryutaro Okuma

Version 0:

Reviewer comments:

Reviewer #1

(Remarks to the Author)

The authors present the successful synthesis of β -Na₂PrO₃, which hosts an underexplored hyperhoneycomb lattice. The magnetic sites are 4f¹ spins of Pr⁴⁺, where strong spin orbit coupling is argued to give rise to bond-dependent interactions that are well described by a quantum compass model. Although the study was inspired by the possibility of Kitaev interactions, the authors argue that the physics are distinct from the Kitaev physics of Ir and Ru 4d and 5d materials.

My overall impression of the work is positive, and I believe that the work is suitable for publication. The results are noteworthy in that a new avenue to anisotropic magnetism, which until now has only been discussed in passing, has been experimentally realized. This may further open the door to other hyperhoneycomb materials, which may or may not be related to Kitaev physics, but indeed seem to feature anisotropic magnetism.

The refinement of the magnetic structure is sound and well written, but I have one question. The lineshape of the magnetic peak in Fig. 3(d) is broad – is it possible to establish a correlation length from this? Is the magnetic structure long-range in nature or does it have a short correlation length as one might expect from a frustrated system?

Were any heat capacity measurements performed at He³ temperatures to get an estimate of the behavior of the spin gap as a function of field? Such an evolution of the spin gap is analytically calculable with linear spin-wave theory, using the model refined in the text.

I would suggest adding a brief reference in the main text to the discussion of the alternative Heisenberg model with DM interactions that is in the SI. The applicability of the anisotropic $J\Gamma\Gamma'$ model is clear from this, but I do not think that it is immediately clear that bond-dependent interactions must be significant in this system, so this discussion is helpful and should be highlighted.

Although it is beyond the scope of this work, I am curious why the Kitaev interaction is completely extinguished, and why the Heisenberg interaction seems to dominate. While the authors briefly mention that it likely due to the orbital character of Pr⁴⁺, before pursuing further synthesis of hyperhoneycomb rare earth materials some ab-initio studies may be useful to determine what kinds of interactions are even possible in f-electron honeycomb/hyperhoneycomb systems. I suspect that the spin model found in this work may be a universal feature of such systems.

Reviewer #2

(Remarks to the Author)

In this work, Okuma and colleagues have conducted a detailed investigation into the magnetic properties of beta-Na₂PrO₃. They have successfully synthesized both powders and single crystals of the compound. Their comprehensive experimental approach, which included neutron diffraction, inelastic neutron scattering, magnetometry, and X-ray diffraction, allowed them to systematically analyze the data and draw conclusions about the material's magnetic structure and the underlying magnetic interactions.

I find most of the paper's conclusions convincing and agreeable. In fact, given the material's structural complexity, the

authors have done a great job finding a self-consistent understanding of their measurement data. The quality of the research is high, and the work will likely be seen as a reliable starting point for further investigations into beta-Na₂PrO₃ and perhaps materials with a similar crystal structure and magnetic characteristics.

However, when considering this manuscript for publication in Nature Communications, I have reservations concerning its broad impact and relevance:

1. The compound has a three-dimensional hyper-honeycomb crystal structure, which comes with a large unit cell and relatively complicated atomic positions and neighboring geometry. The crystallographic symmetry appears to be sufficiently low to allow for seemingly complicated $q=0$ antiferromagnetic structure - the highest site symmetry of the atoms is only C_2 , and none of the atoms are occupying a Wyckoff position with locally maximal symmetry. That is to say, when the low symmetry is combined with the effect of spin-orbit coupling, it is actually not so surprising that non-collinear magnetic order develops at low temperature.

(This is not to say that it is unimpressive that the authors have actually managed to determine the magnetic structure.)

2. The paper currently lacks robust evidence supporting the presence of strong magnetic frustration. The modest difference between the Curie-Weiss and Neel temperatures, along with the absence of notable fluctuations within the spin-wave gap (Fig. 4), weakens the argument for a strongly frustrated magnetic ground state. I suggest the authors consider additional experiments, such as low-temperature μ SR, to explore remaining spin dynamics in the ordered state, which might substantiate the claim of significant frustration.

3. The difficulties associated with synthesizing the beta phase of this material and its sensitivity to air are significant drawbacks. These factors may deter further experimental investigation and limit the broader applicability of the findings. There appears to be no other structurally related rare-earth compounds that can broaden the relevance of the present results either.

Considering these aspects, the manuscript might not meet the impact criteria expected by Nature Communications, though it is undoubtedly solid scientifically. It may find a better fit within a specialized journal focused on magnetic materials.

Additional suggestions:

1) It is emphasized that the material is in the "extreme spin-orbit regime". What does this mean? If J is several times greater than Γ , I feel that one cannot say this. Do the authors have INS data showing the crystal-field (or spin-orbit excitation) levels? If so, it would be useful to compare them with theoretical calculations, in order to evaluate how dominant the spin-orbit effects are over the crystal-field energies.

2) The depiction of the magnetic structure in Figure 3g would benefit from additional views along the a , b , and c axes to better appreciate the alignment of spin directions with the crystal structure. A discussion on the effect of the crystallographic inversion and C_2 rotational symmetries on the magnetic structure, such as in the caption of Supplementary Table VIII, could elucidate potential macroscopic effects.

3) The pronounced dynamic correlations observed near 1.3 \AA^{-1} at the Neel temperature are noteworthy. An expanded theoretical analysis, perhaps through Monte Carlo simulations, would provide a deeper understanding of these features at T_N .

Version 1:

Reviewer comments:

Reviewer #1

(Remarks to the Author)

The authors have addressed all of my questions and I believe that the work is suitable for publication.

Reviewer #2

(Remarks to the Author)

The authors have successfully addressed my concerns with their revised manuscript and response to my comments. In particular, I am happy with the added information regarding how they handled their samples to avoid damage by moisture and ensure appropriate background subtraction, as well as with their new report to be published in Inorganic Chemistry regarding the preparative chemistry. These render the Na₂PrO₃ polymorphs more accessible to in-depth studies, and the extended discussion about the related materials helps connect the present work to a broader readership. The authors' clarifications about the strengths of magnetic frustration and spin-orbit coupling are agreeable, and their modifications of some of the wording in the manuscript helps. I am thus happy to recommend the revised manuscript for publication in Nature Communications.

Below we provide point-by-point responses to the referees' comments. Referees' and our comments are shown in black and blue, respectively. Changes in the revised manuscript are shown in red.

Reviewer #1

The authors present the successful synthesis of β -Na₂PrO₃, which hosts an underexplored hyperhoneycomb lattice. The magnetic sites are 4f1 spins of Pr⁴⁺, where strong spin orbit coupling is argued to give rise to bond-dependent interactions that are well described by a quantum compass model. Although the study was inspired by the possibility of Kitaev interactions, the authors argue that the physics are distinct from the Kitaev physics of Ir and Ru 4d and 5d materials.

My overall impression of the work is positive, and I believe that the work is suitable for publication.

The results are noteworthy in that a new avenue to anisotropic magnetism, which until now has only been discussed in passing, has been experimentally realized. This may further open the door to other hyperhoneycomb materials, which may or may not be related to Kitaev physics, but indeed seem to feature anisotropic magnetism.

We thank the referee for appreciating the importance of our work.

1. The refinement of the magnetic structure is sound and well written, but I have one question. The lineshape of the magnetic peak in Fig. 3(d) is broad – is it possible to establish a correlation length from this? Is the magnetic structure long-range in nature or does it have a short correlation length as one might expect from a frustrated system?

Following the referee's question, we compare in Fig. R1 the magnetic and structural diffraction patterns. The magnetic and structural (022) peak have almost identical lineshapes, which suggests that the correlation length that describes the antiferromagnetically-coupled a-axis moment components (as measured by the (022) reflection) and the structural correlation lengths are the same, i.e. there is no indication from this comparison that the magnetic order is short range, but rather that it has long-range character.

Fig. R1 Magnetic diffraction pattern at 1.4 K (red open circles and red line, reproduced from Fig 3e, obtained as the difference between the 1.4 K and paramagnetic 10 K raw diffraction patterns), compared with the (raw) structural diffraction pattern at 10 K (black open triangles). The intensities in the structural pattern were scaled by $\times 0.06$ and vertically shifted by an offset of -65 to enable the direct comparison of the magnetic and structural (022) peak lineshapes. Note that the (113) structural peak is absent because of the reason discussed in the Supplementary Figure 3.

For completeness we also consider the magnetic (002) peak in Fig 3d (replotted in Fig. R2 below) mentioned by the referee. Unfortunately, there is no close-by structural peak to make a direct comparison, however, with reference to data measured from a high quality $\text{Na}_2\text{Ca}_3\text{Al}_2\text{F}_{14}$ standard one can show that the peak width is predominantly determined by the instrumental resolution, and therefore consistent with the data shown in Figure R1. Significant peak broadening occurs towards lower-angle / higher d-spacing due to instrumental effects (the resolution of a time-of-flight diffractometer has a $\cot^2(\theta)$ dependence, so at the low scattering angles necessary to observe the long d-spacing peaks the resolution is wider and peaks appear broader).

Fig. R2 Fit of the purely-magnetic (002) diffraction peak using an appropriate peak profile in the FullProf Suite (time of flight Convolution pseudo-Voigt with back-to-back exponential function), which allows fitting an intrinsic peak width and extraction of a magnetic correlation length. The experimental data is reproduced from Fig. 3d in the main text.

Based on the discussion above, we have added the following text in the caption of Fig. 3d.

We have verified that the structural and magnetic peaks at similar d -spacing have comparable widths, which indicates that the magnetic order has long-range character. The apparent broader peak shape in Fig. 3d is due to the instrumental resolution becoming broader at high d -spacing.

2. Were any heat capacity measurements performed at He3 temperatures to get an estimate of the behavior of the spin gap as a function of field? Such an evolution of the spin gap is analytically calculable with linear spin-wave theory, using the model refined in the text.

We thank the referee for this insightful suggestion. We have not yet performed the detailed study that the referee is suggesting, but have performed successful preliminary measurements to test the feasibility of heat capacity measurements in high magnetic fields. However, doing a detailed study as a function of temperature, field strength and different field directions (as strong anisotropy is expected), combined with quantitative comparison with a spin wave model would be a rather substantial, self-contained separate project in itself. This is something we are currently pursuing, but we hope the referee appreciates such a study is outside the scope of the present paper.

3. I would suggest adding a brief reference in the main text to the discussion of the alternative Heisenberg model with DM interactions that is in the SI. The applicability of the anisotropic $J \Gamma \Gamma'$ model is clear from this, but I do not think that it is immediately clear that bond-dependent interactions must be significant in this system, so this discussion is helpful and should be highlighted.

In light of the referee's suggestion, we have added the following paragraph in the main text to refer to the "Supplementary Note 5 with comparison to a Heisenberg model with Dzyaloshinskii Moriya (DM) interactions"

We note that the non-collinear magnetic order observed could be stabilized by the competition between just two exchange terms, a nearest-neighbour antiferromagnetic Heisenberg exchange J and a symmetry-allowed Dzyaloshinskii-Moriya interaction D on the vertical bonds. However, to quantitatively explain the large angle between moments in the ab plane would require an unphysically large D/J ratio and also such a model would have a gapless spectrum, in contrast with the substantial gap seen experimentally (for more details see Supplementary Note 5 in [21]). We therefore consider the origin of the observed magnetic structure and spin wave excitations within a JKT model, ...

4. Although it is beyond the scope of this work, I am curious why the Kitaev interaction is completely extinguished, and why the Heisenberg interaction seems to dominate. While the authors briefly mention that it likely due to the orbital character of Pr^{4+} , before pursuing further synthesis of hyperhoneycomb rare earth materials some *ab-initio* studies may be useful to determine what kinds of interactions are even possible in f-electron honeycomb/hyperhoneycomb systems. I suspect that the spin model found in this work may be a universal feature of such systems.

We share the referee's curiosity as to what key factors determine the form of the magnetic exchange interactions and we hope that our synthesis breakthrough of this class of materials, with full magnetic structure solution and proposed quantitative Hamiltonian for hyperhoneycomb $\beta\text{-Na}_2\text{PrO}_3$ will stimulate further *ab initio* theoretical work to understand microscopically the fundamental origin of the different exchange terms in the Hamiltonian of f-electron systems. It may well be that the spin model we found could be a universal feature of many f-electron systems. We note that the original prediction of dominant antiferromagnetic Kitaev interactions in α - and $\beta\text{-Na}_2\text{PrO}_3$ in ref.16 relied on the assumption that the strength of the crystal electric field was negligibly small such that it could be treated perturbatively in relation to the strong atomic spin orbit coupling. However, this may not be the case in the Na_2PrO_3 family and other Pr^{4+} oxides, the crystal electric field may be stronger than one can treat reliably in the perturbative limit as discussed in ref. 18. Very recently, Jang et al. *arXiv*: 2404.17058 (2024) showed that in an alternative coupling scheme the Heisenberg interaction dominates and the Kitaev interaction becomes subleading, so which term dominates depends on the fine balance between spin orbit coupling and the strength of the crystal field. We hope that our detailed structural studies combined with a full magnetic structure solution and minimal Hamiltonian fitted to inelastic neutron scattering data will stimulate more theoretical work in the direction of microscopic understanding of the origin of the different exchange terms that will ultimately help in the search for materials with exotic quantum spin Hamiltonians.

Reviewer #2 (Remarks to the Author):

In this work, Okuma and colleagues have conducted a detailed investigation into the magnetic properties of beta-Na₂PrO₃. They have successfully synthesized both powders and single crystals of the compound. Their comprehensive experimental approach, which included neutron diffraction, inelastic neutron scattering, magnetometry, and X-ray diffraction, allowed them to systematically analyze the data and draw conclusions about the material's magnetic structure and the underlying magnetic interactions.

I find most of the paper's conclusions convincing and agreeable. In fact, given the material's structural complexity, the authors have done a great job finding a self-consistent understanding of their measurement data. The quality of the research is high, and the work will likely be seen as a reliable starting point for further investigations into beta-Na₂PrO₃ and perhaps materials with a similar crystal structure and magnetic characteristics.

We thank the referee for appreciating the quality of our work.

However, when considering this manuscript for publication in Nature Communications, I have reservations concerning its broad impact and relevance:

We address the referee's request for clarification of the broad impact and relevance of our work below.

1. The compound has a three-dimensional hyper-honeycomb crystal structure, which comes with a large unit cell and relatively complicated atomic positions and neighboring geometry. The crystallographic symmetry appears to be sufficiently low to allow for seemingly complicated $q=0$ antiferromagnetic structure - the highest site symmetry of the atoms is only C_2 , and none of the atoms are occupying a Wyckoff position with locally maximal symmetry. That is to say, when the low symmetry is combined with the effect of spin-orbit coupling, it is actually not so surprising that non-collinear magnetic order develops at low temperature.

(This is not to say that it is unimpressive that the authors have actually managed to determine the magnetic structure.)

We thank the referee for appreciating the challenges and effort involved to completely and reliably solve the magnetic structure. Regarding the low local symmetry of the Pr atom, as the referee points out, a more common energy term, the single ion anisotropy produced by the crystal field and spin orbit coupling and constrained by the low local symmetry could in principle lead to non-collinear orders because the local easy axis at the magnetic moment site is allowed to be non-parallel in the ab plane between two zig-zag chains (inversion symmetry between consecutive sites on the same zigzag chain keeps the local easy axis direction the same for all sites in the same family of zigzag chains). However,

this mechanism only applies in systems with large spin ($S > 1/2$), this mechanism is not operational in the present case as no single ion anisotropy can exist because Pr^{4+} is a Kramers ion with an odd number of electrons and in this case the magnetic ground state is a doublet described by an effective spin-1/2. Considering a single site the magnetic energy is completely independent of the moment orientation, there is no locally-preferred direction, all directions are energetically degenerate by virtue of the spin-1/2 operator properties (the expectation value of every squared spin component along any direction is a constant $\langle S_x^2 \rangle = \langle S_y^2 \rangle = \langle S_z^2 \rangle = 1/4$ independent of what wavefunction the spin is in, and the local anisotropy is in general a polynomial of even powers of the spin components, so it is a constant). The presence of a spin gap and of a non-collinear magnetic order are non-trivial effects in the sense that those are not due to single ion physics, both effects are a direct manifestation of the cooperative behaviour of all spins acting together as a consequence of the bond-dependent interactions between them. Local physics in terms of the on-site symmetry and spin orbit coupling cannot explain the observed non-collinear magnetic structure or the presence of the gap, bond-dependent interactions between spins are needed to explain those effects. This is the very reason why we invoked the frustrated J - Γ model with substantial bond dependent Γ terms. It is the frustration between the easy axes selected by the Γ terms on the three different types of bonds that stabilizes the non-collinear ground state.

2. The paper currently lacks robust evidence supporting the presence of strong magnetic frustration. The modest difference between the Curie-Weiss and Neel temperatures, along with the absence of notable fluctuations within the spin-wave gap (Fig. 4), weakens the argument for a strongly frustrated magnetic ground state. I suggest the authors consider additional experiments, such as low-temperature μSR , to explore remaining spin dynamics in the ordered state, which might substantiate the claim of significant frustration.

We of course agree with the referee that the system is not as strongly frustrated as the Kitaev model, and to avoid any confusion in the revised manuscript we have removed the word “strong” from in front of “frustration” in a couple of instances where it was mentioned. However, frustration is key to the observed physics, the interactions are frustrated as the bond-dependent Γ exchanges on the three bonds meeting at every site prefer incompatible easy axes. The system is frustrated as no spin arrangement can simultaneously fully satisfy all interactions and the non-collinear order we observe is the resulting compromise. We thank the referee for their insightful suggestion of performing μSR measurements to further test the dipolar fields in the ordered phase, those would indeed provide additional useful information, but we feel such a study is outside the immediate scope of the present manuscript. The present datasets already provide, we believe, compelling experimental evidence of a novel type of frustrated interaction, the Γ term, in the spin Hamiltonian. To the best of our knowledge the material under study is the first robust realization of such an interaction on a threefold coordinated lattice and we hope our results will stimulate future experimental and theoretical developments of new physics based on the Γ interaction beyond the mostly explored Kitaev physics.

3. The difficulties associated with synthesizing the beta phase of this material and its sensitivity to air are significant drawbacks. These factors may deter further experimental investigation and limit the broader applicability of the findings. There appears to be no other structurally related rare-earth compounds that can broaden the relevance of the present results either.

As the referee points out, **up to now** the moisture-sensitivity of the compound and starting chemicals severely limited the access to experimental investigation. However, in the present manuscript and in the attached report accepted in *Inorganic Chemistry*, we made a breakthrough in understanding the detailed chemistry, presented a thorough investigation of the preparative chemistry of Na-Pr-O phases and firmly established the conditions in which powder and single crystals of honeycomb and hyperhoneycomb Na_2PrO_3 can be reliably and reproducibly synthesized. Furthermore, once the compound is crystallized, detailed physical properties can be revealed as we show by the various measurements performed in the manuscript, and preliminary high field heat capacity. We believe that our comprehensive chemical and physical characterization of the Na_2PrO_3 family will encourage the community to develop and investigate quantum materials that have been overlooked up to now due to synthetic or handling difficulties. We note that the Nature family of journals has over the years published many papers on quantum materials that required inert atmosphere handling or synthesis conditions, for example FeI_2 and CoI_2 [Bai, X., Zhang, S. S., Dun, et al. Hybridized quadrupolar excitations in the spin-anisotropic frustrated magnet FeI_2 . *Nat. Phys.*, 17, 467-472 (2021); Kim, C., Kim, S., Park, P. et al. Bond-dependent anisotropy and magnon decay in cobalt-based Kitaev triangular antiferromagnet. *Nat. Phys.* 19, 1624–1629 (2023).]. We hope that the referee agrees that if we are to only investigate materials that are not air sensitive, then we are closing a lot of opportunities to discover new science.

The following text has been added to the Methods section with tips about sample handling.

The single crystals were initially handled in vacuum pump oil, which was removed by washing with toluene. For the magnetization measurements in field along specific crystallographic directions (*a*, *b*, *c*) the crystals were aligned and fixed onto a flat plate (single crystal of NaCl with typical dimensions $1.4 \times 1.3 \times 0.7 \text{ mm}^3$). Molten paraffin wax was used to coat the crystals and fix them onto the flat plate and an aluminium foil was attached below the flat plate to roughly cancel the large diamagnetic contribution of paraffin wax. After the measurements along all the axes, the crystals were removed by melting the wax and the background susceptibility signal, comprising of diamagnetism from wax and NaCl, and paramagnetism of the aluminium foil, was measured and subtracted off to obtain the intrinsic Na_2PrO_3 signal.

Experimentally, there is an ever increasing number of new materials with three-dimensional threefold coordinated lattices such as the hyperhoneycomb being discovered, and they are invariably found to be of considerable interest For example, IrI_3 , which forms a two-dimensional honeycomb lattice of Ir^{3+}

ions when synthesized at ambient pressure, becomes a hyperhoneycomb lattice when synthesized under high pressure (*J. Solid State Chem.* 312, 123240 (2022).). Although Ir³⁺ ions are non-magnetic, this work is remarkable as it demonstrates that the hyperhoneycomb lattice geometry can be realized even in non-oxide materials. Very recently, another three-dimensional threefold coordinated lattice, a hyperoctagon lattice of $j_{\text{eff}} = 1/2$ ions has been synthesized as a Co-based metal organic framework (*Phys. Rev. Lett.* 132, 156702 (2024).). Beyond threefold coordinated lattices, some ordered rock-salt structures have three-dimensional edge-shared octahedra and have the same crystallographic space group (Fddd) as that of the hyperhoneycomb lattice (*Inorg. Chem.* 58, 13881 (2019).; *Inorg. Chem.* 61, 10880 (2022).). These materials await in-depth physical characterizations for new physics as we have done in the current work. Based on the discussion above, we have added a new end paragraph in the Discussion.

Beyond β,γ -Li₂IrO₃ and β -Na₂PrO₃, realization of three-dimensional threefold coordinated lattices in more systems will be an important task to reveal the rich physics of the quantum compass models. Notable progress in this direction is the recently demonstrated control of cation ordering in the rock salt structure, used to obtain a three-dimensional network of corner- and edge-sharing octahedra in Li₃Co₂SbO₆ [31,32]. The use of high pressure could also potentially transform layered honeycomb materials into 3D hyperhoneycomb lattices, as illustrated by high-pressure studies on IrI₃ [33]. Very recently, an organic molecule was employed to realize another threefold coordinated lattice of much theoretical interest, the hyperoctagon lattice, in a Co-based metal organic framework [34]. We anticipate that these and other novel synthetic procedures will expand the materials platform of three-dimensional threefold coordinated lattices and allow a wider experimental exploration of the rich range of cooperative magnetic behaviours expected for such geometries in the presence of strong spin orbit coupling, as we have revealed in hyperhoneycomb β -Na₂PrO₃.

Considering these aspects, the manuscript might not meet the impact criteria expected by Nature Communications, though it is undoubtedly solid scientifically. It may find a better fit within a specialized journal focused on magnetic materials.

We hope that our above responses have provided justification that our manuscript does meet the broad impact criteria expected by Nature Communications.

Additional suggestions:

1) It is emphasized that the material is in the "extreme spin-orbit regime". What does this mean? If J is several times greater than Γ , I feel that one cannot say this. Do the authors have INS data showing the crystal-field (or spin-orbit excitation) levels? If so, it would be useful to compare them with theoretical calculations, in order to evaluate how dominant the spin-orbit effects are over the crystal-field energies.

We fully agree with the referee that performing a detailed study of the crystal-field levels would add further information to characterize the strength of the crystal field terms, the relevance of the cubic versus non-cubic terms, and provide further insight into the origin of the observed reduced g-factor. However, we feel that such a study is beyond the scope of the present manuscript, the availability of results of such a study are not crucial for the main conclusions reported in the present manuscript and we plan to leave such a study for future work.

The referee asked for clarification on the use of the terminology “extreme spin orbit regime”. We use this terminology to draw a distinction from the Ir⁴⁺ oxides (iridates), described to be in the “strong spin-orbit coupling limit” [G. Jackeli, and G. Khaliullin, Mott Insulators in the Strong Spin-Orbit Coupling Limit: From Heisenberg to a Quantum Compass and Kitaev Models, PRL 102, 017205 (2009)]. In particular, for the rare earth Pr⁴⁺ oxides the spin orbit coupling λ is expected to be larger than (ref [15]), or at least comparable to [Jang et al. *arXiv*: 2404.17058 (2024)], the dominant octahedral cubic crystal field Δ . This is in contrast to Ir⁴⁺ oxides, where the spin orbit coupling is significantly smaller ($\lambda \sim 0.4$ eV) than the dominant (octahedral cubic) crystal field ($\Delta \sim 3$ eV), but is larger than sub-leading, non-cubic crystal field terms [PRL 102, 017205 (2009)]. For cubic crystal field, a spin-orbital entangled $j_{\text{eff}}=1/2$ state is stabilized in both cases, but the microscopic origins and wavefunctions are different, so we feel it is appropriate to emphasize this distinction by using a terminology like “very strong” or “extreme” spin orbit regime and highlight that Pr⁴⁺ oxides offer a distinct materials platform to explore the cooperative physics of spin-orbit entangled moments coupled by bond-dependent anisotropic exchanges. The presence of the Gamma exchange in the spin Hamiltonian indicates that the spin orbit coupling is very relevant even for the superexchange processes, and the fact that the gap (which originates directly from the Gamma terms) is half the dispersive bandwidth indicates that the Gamma terms very substantially affect the dynamics. The referee makes an interesting suggestion to use the ratio of J over Gamma as a way to quantify the relevance of the spin orbit couplings, at the moment we feel inclined to suggest that this single ratio may not be capturing the complete picture, i.e. the spin orbit coupling is essential in stabilizing the magnetic moments with admixed spin-orbit character (and reduced g-factor compared to the spin-only value of 2) in the first place, is influencing the exchange interactions and is affecting very quantitatively the spin dynamics.

2) The depiction of the magnetic structure in Figure 3g would benefit from additional views along the a, b, and c axes to better appreciate the alignment of spin directions with the crystal structure. A discussion on the effect of the crystallographic inversion and C2 rotational symmetries on the magnetic structure, such as in the caption of Supplementary Table VIII, could elucidate potential macroscopic effects.

Following the referee's suggestion we have used this opportunity of the resubmission to add a new Supplementary Figure 6 with views of the magnetic structure along the a , b , c and other axes to better appreciate the alignment of the spins in relation to the crystal axes, and we also re-stated in that figure caption all relevant symmetry operations that leave the magnetic structure invariant. The magnetic space group $Fd'dd$ is mentioned in the Supplementary Table VIII and discussed in more detail in Supplementary Note 3 (Symmetry of the magnetic diffraction pattern).

3) The pronounced dynamic correlations observed near 1.3 \AA^{-1} at the Neel temperature are noteworthy. An expanded theoretical analysis, perhaps through Monte Carlo simulations, would provide a deeper understanding of these features at T_N .

Following the referee's suggestion we have added a new sentence in the revised manuscript to explain the significance of the pronounced dynamical correlations observed at the Neel temperature in Fig. 4c near 1.3 \AA^{-1} . We note that in this $|Q|$ -region several strong magnetic Bragg peaks are present in the ordered phase (blue vertical bars at the bottom of Fig 4a) and it is at those wavevectors that the dispersions soften as well, where the minimum gap positions are observed in Fig. 4a at base temperature. Therefore, it is quite natural that dynamical correlations still persist at those wavevectors at the Neel temperature and just above, it is from those dynamical correlations that the magnetic order develops upon cooling below the Neel temperature. Added text to Fig. 4d caption:

The enhanced inelastic signal in a broad momentum range near 1.3 \AA^{-1} and extending down to the lowest energies is attributed to precursor dynamical correlations from which the magnetic order develops upon cooling.